# Does the Therapist’s Sex Affect the Psychological Effects of Sports Massage?—A Quasi-Experimental Study

**DOI:** 10.3390/brainsci10060376

**Published:** 2020-06-16

**Authors:** Bernhard Reichert

**Affiliations:** 1Institute for Circulatory Research and Sports Medicine, German Sports University Cologne, Am Sportpark Müngersdorf 6, 50933 Köln, Germany; bernhard.reichert@stud.dshs-koeln.de or info@di-uni.de or mail@bernhardreichert.de; 2Public Health and Medicine, Dresden International University, Freiberger. Str. 37, 01067 Dresden, Germany

**Keywords:** sports massage, current emotional state, mood, therapist’s sex, athlete’s sex

## Abstract

**Objectives**: The aim of this study was to determine the influence of the sex of the therapist and of the athlete on the athlete’s current emotional state after a sports massage. The assumption was that the effect of a massage on the current mood was independent of the sex of the therapists or athletes. **Background**: Sports massages are an integral part of the support given to athletes during training or competition and are a commonly used method for promoting athletes’ physical and mental recovery. Few studies have measured the mental characteristics or even the nonspecific effects of sports massages. Sexual attraction or dislike are among the nonspecific effects of a treatment. **Materials and methods**: One hundred and sixty-eight high-performance male and female amateur athletes received a sports massage from 15 male and female trained therapists. The current emotional state of the athletes was measured before and after intervention using the BSKE-EA17 adjective scale, whose items can be assigned to five categories of the current emotional state. ANOVAs (analysis of covariances) were carried out to calculate the interactions between the sexes. Cohen’s d for similar group sizes and similar group variances were determined. **Results**: Neither the sex of the therapist nor the sex of the athlete had any influence on the mental effect of a sports massage. The only exception was when male athletes were treated by female therapists, where an increase in “elevated mood” was observed. Sports massages resulted in an increase in the responses in the categories “elevated mood” (d = 1.1) and “level of activation” (d = 0.3) and a decrease in the responses for “low mood” (d = 0.3), “level of deactivation” (d = 0.6) and “level of excitation” after the massage compared to before the massage (d = 0.9). **Conclusions**: Sports massages appear to increase the positive dimensions of the athletes’ current emotional state and reduce the negative dimensions. The self-reported mood changes from before the massage to after the massage were not influenced by other prognostic variables, including wait time, age of the athlete or the duration of the run. The results suggest that the specific effects of sports massages on the mental status are supported. Disregarding the aspect of the therapists’ sex, sports officials, trainers and athletes therefore can be more independent in the personnel planning of sports therapists.

## 1. Introduction

Massage is the manipulation of soft tissue by a trained therapist as a component of a holistic therapeutic intervention [1,2].

Sports massage is defined as a set of massage techniques that enhance athletes’ recovery and help treat pathological conditions [3]. Depending on the time a sports massage is performed in relation to exercise, the terms “pre-event”, “pre-exercise”, “inter-event”, “post-event”, “post-exercise” and “training” massage are used [4].

The specific forms of a massage are broken down further into the overall duration, intensity and selection of techniques used. In most studies, technical elements of Swedish, classic or Western massage are used in sports massages: strokes, kneading, petrissage, frictions and vibrations. In the literature, the term “recovery massage” is used to describe a massage delivered after intense exercise [4]. The best outcomes of massages for muscular recovery were achieved when the treatment was applied within the first two hours of exercise [5].

The particular significance of a recovery massage is especially evident in athletes performing repeated exertions within a short period. In athletes competing in multidiscipline events, as well as those participating in various disciplines, top performances must be delivered multiple times on the same day or several days in a row. For this reason, a quick recovery is an important factor for ensuring top performances for the duration of the competition [1].

A sports massage plays a valuable role in the health system [6]. It is an integral component of support for athletes during training or competition. Athletes, those supporting them and individuals associated with sports worldwide recognize sports massages as an effective means of boosting recovery and reducing pain and discomfort [3]. Sports massages are often offered to support athletes at large-scale events. Galloway and Watt reported in 2004 that physical therapists devoted 24.0% to 52.2% of the time they spent supporting athletes at large national and international sports events to massages [7].

Scientific aspects of sports massages are of interest to athletes, their trainers and sports physicians [8]. Schilz and Leach (2020) surveyed 100 endurance athletes about their knowledge regarding the effects of sports massage therapy. Of the athletes surveyed, 93% felt that sports massage therapy could be seen as a form of injury prevention, 92% felt that it was a valuable method of resolving a wide variety of muscular problems and 90% indicated that they felt that sports massage therapy improved their quality of life.

Very few studies have addressed sports massages following endurance performances [9] and very few address psychological factors [9,10,11]. The latest relevant review by Poppendieck (2016) described the benefits of post-exercise sports massages in detail. These positive effects were observed for strength and endurance exercises, as well as for high-intensity mixed exercises. Since the physiological mechanisms of the performance recovery were unclear for the authors, they described the mental effects as more significant [12].

In general, the study objects, outcomes and results of the different (physiological, in particular) effects of sports massages vary widely, and unequivocal statements are rare [8,13]. Very few studies have measured mental characteristics.

Each therapeutic intervention can be assumed to have an effect component that is directly dependent on the intervention (specific effect) and an effect component that is independent of the intervention (nonspecific effect) [14]. These nonspecific effects include the presence, voice and sex of the therapist; trust; therapeutic alliance [15] and space-related conditions. In order to better classify the value of massage therapy (specific effects), the influences of nonspecific effects on, e.g., mental outcomes should be investigated. No studies are available in the well-known medical databases that investigate the nonspecific effects of a massage with respect to psychological effects [14].

Experience has shown that athletes rarely perceive physical contact occurring during manually applied massages to be unpleasant or anticipate that it will be unpleasant [16]. During the application, one could assume that massage therapy may result in an improved mental status, depending on the sex of the therapist or the athlete, through some kind of sexual stimulation. Culturally formed attitudes may prohibit heterosexual contact between the therapist and the athlete. Negative experiences with heterosexual contact in the personal history of athletes could also lead to preferring a certain sex of the therapist. Therefore, one question that may arise is whether the sex of the therapist or the athlete can contribute to a change in the perception and, thus, contribute to the athlete’s assessment of his or her current emotional state. This possible influence has not been investigated in previous studies to date.

The aim of this study was therefore to determine the influence of the sex of the therapist and of the athlete on the athlete’s current mental state after a sports massage. The assumption is that the effects of a massage on the current mood is independent of the sex of therapists or athletes. The following research questions led through the study:(1)In general, does the wellbeing of the athlete change from before the intervention to after the intervention (main effect of time)?(2)Does the sex of the therapist have a (main) influence on the change in the wellbeing of the athlete?(3)Does the change of wellbeing depend on the sex of the athlete (so-called interaction effect or interaction)?

## 2. Materials and Methods

### 2.1. Participants and Setting

The participants in this quasi-experimental study design were recruited on the day prior to the 19th Stuttgart half-marathon in the city of Stuttgart, Germany. Some 19,000 athletes participated in the race. All of the study participants were ambitious amateur athletes. They were personally approached and encouraged to participate in the study. Recruiting took place in the Hanns-Martin-Schleyer-Halle, a large sports and entertainment venue in Stuttgart. The participants were informed in writing, gave their consent after having read the information and completed a questionnaire to clarify the inclusion/exclusion criteria.

Included in the study were participants in the half-marathon who were between the ages of 18 and 70 and who did not meet any of the exclusion criteria.

Any health condition that negatively impacted the athletes’ overall performance capacity—in particular, their running performance, their recovery and the perception of tactile stimuli—were defined as exclusion criteria. They included the following medical criteria:Musculoskeletal system: Disorders of or injuries to muscles, joints or vertebral disks, as well as artificial hip, knee or ankle replacements.Nervous system: Disorders such as polyneuropathy, multiple sclerosis or paralysis.Cardiovascular system: Elevated blood pressure, antihypertensive drugs, heart disease such as arrhythmia, heart failure, heart valve disease, pericarditis or myocarditis.Lungs and respiratory tract: Pulmonary and respiratory disorders and asthma sprays.Renal and metabolism: Renal disorders, kidney transplant and diabetes.

The following variables were recorded as additional prognostic factors:Use of pain-relief medication after the half-marathon.Expected length of the sports massage.Estimated actual length of the sports massage.Influence of the wait time on the current emotional state.Running performance on the race day and weekly training time in the previous three months.

#### 2.1.1. Sample size Calculation

The maximum number of participants was determined based on the number of therapists delivering massages simultaneously (15) and the average duration of the massage therapy. All therapists gave massages continuously and treated as long as there was demand from athletes. All included test persons who visited the massage area were treated.

#### 2.1.2. Assignment of Participants and Blinding

Whenever a massage table freed up, a new athlete was assigned to it, regardless of the sex of either the athlete or the therapist. This means that the male and female athletes were assigned to the male and female therapists in a random fashion, albeit not in the conventional sense. Both male and female athletes participating in the study and male and female athletes not participating in the study underwent massage in the same setting. The participants knew that they were part of a study. The subjects were instructed not to tell the therapists that they were participating in the study, so the therapists were unable to identify these athletes as study participants. The athletes were assigned an ID number, so that the individuals evaluating the questionnaires and those analyzing the data were blinded.

### 2.2. Study Procedure and Intervention

The participants were requested to come to the massage area on the day of the half-marathon as soon as possible after crossing the finish line. Immediately after they had checked in and before the massage, they filled out a sociodemographic questionnaire and completed a questionnaire about their current emotional state. Immediately after the massage, they completed a second questionnaire about their current emotional state. In terms of the mental outcome variables, there were no differences between the pre-massage and post-massage questionnaires. The participants completed the questionnaires in a separate waiting area within the massage area.

The massages were delivered near the finish area of the half-marathon in a separate area in which 15 massage tables were set up next to each other without any partitions. Participants and athletes not participating in the study were treated identically and were assigned to a free massage table.

The massage therapists (8 male and 7 female) were students enrolled in the physical therapy school in Fellbach, Germany and were trained for several hours to perform the sports massage standardized in duration, techniques, sequence and intensity. The massage length was set at 15 to 20 min. The massage entailed the treatment of two legs only with the techniques used in classic massage therapy: strokes, kneading, petrissage and frictions, with the balls of the hands on the front and back of the legs. This corresponds to the way a sports massage is performed on endurance athletes [16] and the way sports massages were conducted in previous studies by the author [17]. A neutral commercially available massage oil without any additives was used for the massage. All of the sports massages took place on the same day between approximately 11:00 a.m. and 4:00 p.m.

### 2.3. Measurement Instruments

In addition to a sociodemographic questionnaire with additional prognostic variables, a questionnaire for describing the athlete’s current mental state was used. The latter questionnaire comprised a short form of the adjective list compiled by Janke, Erdmann and Hüppe (BSKE-EA 17), which was used in various previous studies [18]. This 17-item scale was suitable for recording the respondents’ current emotional states and any short-term changes. The items were assigned to different categories:Elevated mood, three items: emotional wellbeing, feeling of being relaxed and feeling of joyLow mood, five items: dysphoric feeling, feeling anxious, feeling sad, feeling angry and feeling physically unwellLevel of activation, two items: feeling of being active and feeling of alertnessDeactivation, two items: lack of energy and feeling tiredLevel of excitement, four items: feeling of inner excitement, feeling of physical excitement, feeling of shakiness and feeling of inner tension

Mood in psychology is a form of pleasant or unpleasant feeling that forms the background of human experiences. Mood depends on the (biological) overall constitution of the individual and the individual’s current emotional state [19].

Janke and Debus described some of these categories in their manual [20] as follows:

Activation: “State-of-mind feature characterized by pleasure-oriented activity that is primarily performance-oriented but also environmentally oriented [...] and encompasses holistically somatic and psychological aspects of activity.” A high rating for activation describes a state of the highest possible performance efficiency and wellbeing.

Deactivation: “State-of-mind feature characterized by reduced activity with respect to performance and the environment (relation to introversion!). This reduced activity is closely connected to the feeling of overall impaired willingness and ability to perform in the sense that the individual feels a lack of ability and lack of willingness to do anything.”

Excitement: “State-of-mind feature characterized by motor restlessness and tension characterized by lack of desire and emotional disequilibrium (emotional lability), combined with performance inefficiency.”

Each description of the current emotional state (item) could be evaluated using a Likert scale in seven gradations. These items were coded between 0 = “not at all” and 6 = “very strong”. The participants used this instrument to evaluate themselves and indicate to what extent certain feeling states corresponded to their current emotional states. Each scale is marked by a noun and two exemplary adjectives (see example in Figure 1).

This yields the following point ranges for each category: elevated mood (0 to 18 points), low mood (0 to 30 points), activation (0 to 12 points), deactivation (0 to 12 points) and excitement (0 to 24 points). Totaling the scores of these scales would not be expedient, which is why the items were evaluated by category.

### 2.4. Objectives and Outcomes

The primary outcome of this study was to describe the influence of the therapist’s sex and of the participant’s sex on the psychological effects of a sports massage. It was assumed that neither the sex of the therapist nor that of the athlete would have an effect.

The secondary research question was as follows: What general effect does a sports massage have on an athlete’s current state of mind? Do any other prognostic factors (wait time until the massage, age of the athlete or duration of the race) have an influence?

A positive influence on the part of the sports massage on the athlete’s current state of mind is considered to be an increase in the responses in the categories “elevated mood” and “activation” and a decrease in the responses for “low mood”, “deactivation” and “excitement”.

### 2.5. Statistical Methods

The data were recorded in MS Excel [21] and prepared for analysis in IBM SPSS 19.0 [22]. A questionnaire was excluded if more than one item of the outcome variables was not filled out. All other missing values were replaced by mean values. Calculating a total score for the multidimensional emotional state questionnaire (BSKE_EA 17) would not be expedient, which is why the items were evaluated by category. The Kolmogorov-Smirnov test was used to test the metric variables for normal distribution.

*T*-tests for dependent samples were calculated to statistically test the before-and-after differences. The differences in effects between men and women were calculated using the *t*-test for independent samples. To analyze the effects of the sex of the therapist and the effects of the sex of the athlete on the change in the five scales of T1 (pre-massage) to T2 (post-massage), the effect size for mean differences (Cohens d) between two groups (before and after the sports massage) was calculated with similar group sizes and similar group variances. The interpretation of the effect size was based on [23]: low effect size = 0.1 to 0.2, medium effect size = 0.3 to 0.5 and large effect size >0.5.

Mixed ANOVAs were conducted to evaluate the effects of the therapist’s sex and the athlete’s sex on the changes in the five dependent categories (scales). Furthermore, ANOVAs (analysis of covariances) were carried out. In this context, the following hypotheses were tested:(1)The sex of the therapist has a (main) effect on the change in wellbeing.(2)This effect is dependent on the sex of the athlete (“interaction effect”).(3)In general, wellbeing will change from T1 to T2 (main effect of time).

To study the influence of the interval-scaled variables “wait time”, “running performance” (in minutes) and “age” on the change from T1 to T2, multiple regression analyses were performed with the difference values of T2 and T1 as dependent variables. This thus allowed us to investigate whether, for example, shorter wait times were associated with higher difference values (= stronger changes from before to after the massages).

## 3. Results

### 3.1. Deviations from the Protocol

No motivated participants were excluded for health-related reasons before inclusion in the study. No participants discontinued their participation in the study, and no adverse effects of the massages were observed.

### 3.2. Participants’ Characteristics

In all, 200 athletes were recruited—of whom, 185 athletes were included in the study and underwent a sports massage as planned. Seventeen questionnaires were excluded from evaluation owing to missing responses. The responses of 127 male athletes and 41 female athletes were evaluated. Table 1 shows the athletes’ characteristics. They were, on average, 37 years old, approx. 178 cm tall and had a BMI of approx. 23. They had trained for an average of 32 km a week in the previous three months and achieved an average race time of 1:50:55 (h:mm:ss) for the half-marathon distance on the race day. The participants estimated the optimum time taken for a sports massage after a half-marathon at an average of 21 min and estimated the perceived duration of the massage they underwent at 12 min. They waited six minutes on average from the time they entered the massage area to the onset of the sports massage (see Table 1). Seventy-two point two percent reported that the wait time prior to the onset of the sports massage did not negatively influence their current emotional state. Nine athletes (5.7%) took medication prior to the competition that could have an influence on their recovery and current state (e.g., aspirin or diclofenac). The participants received 80 massages by male therapists and 88 massages by female therapists.

### 3.3. Results of the Outcome Variables

Table 2 shows the descriptive analysis for the outcome variables or the before-after comparisons differentiated by group. Sports massages resulted in an increase in the responses in the categories “elevated mood” (d = 1.1) and “activation” (d = 0.3) and a decrease in the responses for “low mood” (d = 0.4), “deactivation” (d = 0.6) and “excitation” after the massage compared to before the massage (d = 0.9). Testing the differences in the results of women and men before and after applications with the two-tailed *t*-test for independent samples revealed a *p*-value <0.05 only for the category “excitement” after the treatment. Table 3 presents the results of the analytical statistics for the outcome variables for the before-after comparisons, differentiated by group.

ANOVAs were conducted to evaluate the effects of the therapist’s sex and the athlete’s sex on the changes in the five dependent categories (scales) between T1 and T2 (Table 4). The resulting design was a 2 (therapist’s sex: male vs. female) × 2 (athlete’s sex: male vs. female) × 2 (time: T1 vs. T2) mixed factorial, with time as a repeated measures factor.

Some comments on some dependent categories:ActivationOf all the other effects, only the main effect of sex of the athlete was significant, *F*(1.163) = 4.4, *p* = 0.04 and η_p_^2^ = 0.03, indicating that males reported overall higher levels of “activation” than females.DeactivationPaired post-hoc (LSD) comparisons showed that the decrease in “deactivation” was significant for almost all combinations of the sex of the athletes and sex of the therapists, all *p* < 0.01, but not for female athletes who received a massage from a male therapist (*p* = 0.59).ExcitementPaired post-hoc (LSD) comparisons showed that the male and female athletes did not differ in their levels of “excitement” before the massage, *p* = 0.36, but after the massage, male athletes reported significantly higher levels of “excitement” than female athletes, *p* < 0.01, albeit lower than before the massage.Effects of wait time, athlete’s age and duration of the runTo analyze the effects of wait time, age of the athlete and duration of the run (time in minutes), regression analyses were conducted with these variables, as prognostic variables and difference scores of the questionnaire scales before and after the massage were used as dependent variables. None of the five regression analyses yielded a significant effect of any of the three prognostic variables.

The hypotheses set out above can be answered as follows:
**Hypothesis** **1:**In general, the current mood changed from T1 to T2 (main effect of time). This hypothesis can be accepted. The low mood decreased significantly from the time before to the time after the massage, regardless of the sex of the therapist or athlete. The feeling of activation increased significantly from the time before to the time after the massage and was greater in men than in women, regardless of the sex of the therapist or athlete. In summary, almost all athletes showed a significant decrease in the feeling of deactivation from the time before to the time after the massage, with the exception of the group of female athletes who were massaged by male therapists. In this group, the feeling of deactivation after the massage was no different. The level of arousal decreased significantly as a result of the sports massage, although male athletes reported significantly higher “arousal levels” than female athletes, although lower than before the massage.
**Hypothesis** **2:**The sex of the therapist had a (main) effect on changes in the current mood. This hypothesis must generally be rejected.
**Hypothesis** **3:**The change in the current mood depended on the sex of the athlete (“interaction effect”). This hypothesis must generally be rejected. The only relevant difference found in the study regarding the influence of the sex of the therapist or the athlete was that both before and after the massage, male athletes reported significantly better moods than female athletes when the male athletes received a massage from a female therapist. All other interactions between the sex of the therapist and the sex of the athlete were not significant.

## 4. Discussion

A total of 185 recruited athletes in the study—of whom, the data of 168 ambitious amateur athletes were evaluated. Fifteen trained students performed the post-exercise massages after a half-marathon. The participant characteristics data corresponded to a survey conducted in 2011 that recorded the data at the same location as the same event and with the same intervention [17].

The aim of this research was to show whether the sex of the therapist or the athlete had an influence on the mental effects of a sports massage and whether the mental state improved with the treatment. In summary, the athletes reported a significantly higher positive and decreased negative mental state after the massage than prior to the massage, and this effect was, in general, identical regardless of the sex of either the therapist or the athlete. The only exception was when male athletes were treated by female therapists, where an increase in “elevated mood” was observed. These results are nearly all congruent with the results of studies addressing the mental recovery of athletes undergoing post-exercise sports massages [9,10,11,16,17]. Hemmings et al. (2000) employed a pre-/post-design to study the physiological and psychological effects of recovery massages in boxers. They showed that the boxers reported a significantly increased perception of recovery after a massage than after passive resting [11]. In a randomized, controlled design involving 108 half-marathon participants, Reichert, 2011 showed that a sports massage significantly increased the elevated mood and reduced the low mood compared to passive resting [17].

In general, too few studies have been conducted with mental outcomes to attest to a fundamental unequivocal effect of sports massages, regardless of athletic discipline. The present study showed that the influence of sex plays, at most, a negligible role with regards to short-term mental outcomes.

## 5. Conclusions

To our knowledge, this was the first study of the nonspecific effects of the sex of therapists and athletes during a massage treatment on their mental status. The study showed that neither the therapist’s sex nor the athlete’s sex influenced the mental effects of the sports massage. Sports massages appear to increase the positive dimensions of the athletes’ current emotional states and reduce the negative dimensions. The self-reported mood changes from before the massages to after the massages were not influenced by other prognostic variables, including the wait time, age of the athlete or the duration of the run. The results suggest supporting the specific effects of sports massages on the mental status. The question of sex is clearly irrelevant to the outcome. Sports officials, trainers and athletes can therefore be more independent in the personnel planning of sports therapists.

In the future, additional studies with a similar study design should be carried out that include control groups and focus on other sports disciplines, as well as on professional athletes. The influence of the therapist’s/athlete’s sex on the mental outcomes should also be investigated for other types of massage (e.g., classic massage) and treatment settings (e.g., rehabilitation). Furthermore, the other nonspecific effects of treatments on these outcomes must also be examined.

## 6. Limitations

Cohen’s d was calculated in the study because no comparable data for classifying the effect size was available from previous studies. For the differences between T1 (pre-massage) and T2 (post-sports massage), however, it should be noted that the lack of a control group meant that, principally, no causal interpretation was possible that would allow the difference to be attributed to the massage. However, in a similar study with a randomized controlled design by Reichert (2011), a significant advantage of sports massage vs. passive rest with regards to the current mental state was suggested [17].

## Figures and Tables

**Figure 1 brainsci-10-00376-f001:**
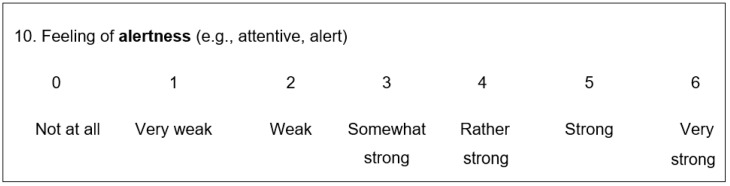
Example of an item from the BSKE-EA 17 scale with gradation and coding.

**Table 1 brainsci-10-00376-t001:** Participant characteristics. MV = mean value, Med = median, SD = standard deviation, Min = minimum, Max = maximum and BMI = body mass index.

Characteristic	MV	Med	SD	Min	Max
**Age** (in years)/all	37.10	37.00	±10.16	18	69
male	37.75	37.00	±10.27	18	69
female	35.10	35.00	±9.56	18	59
**BMI**/all	23.11	22.99	±2.51	18.00	30.87
male	23.72	23.55	±2.35	17.65	30.87
female	21.19	21.05	±1.99	15.22	25.71
**Running time** (h:mm:ss)/all	01:50:55	01:49:00	00:18:20	01:01:00	02:57:00
male	01:47:57	01:48:00	00:17:07	01:19:00	02:57:00
female	02:00:50	02:03:00	00:18:45	01:01:00	02:36:00
**Training** (km)/all	31.91	25.00	±37.25	0	350
male	34.67	30.00	±41.81	0	350
female	23.35	25.00	±13.16	3	50
**Estimated Massage duration** (received, in min)/all	11.94	10.00	±4.10	1	20
male	12.01	10.00	±4.22	1	20
female	11.73	10.00	±3.69	5	20
**Estimated Wait time** (in min)	5.86	5.00	±9.55	0	78
male	6.55	5.00	±1.,76	0	78
female	3.73	3.00	±3.04	0	15.00

**Table 2 brainsci-10-00376-t002:** Descriptive data for the outcome variables. Mean value (standard deviation) and Diff = difference.

Category	Before/All	Bevor/Male	Bevor/Female	After/All	After/Male	After/Female	Diff/All	Diff/Male	Diff/Female
**Elevated mood**	10.88 (±2.49)	10.99 (±2.48)	10.51 (±2.50)	13.63 (±2.45)	13.71 (±2.47)	13.39 (±2.37)	2.76 (±2.58)	2.72 (±2.70)	2.88 (±2.17)
**Low mood**	3.44 (±3.94)	3.71 (±4.27)	2.61 (±2.46)	2.14 (±3.61)	2.31 (±3.85)	1.61 (±2.67)	1.30 (±3.00)	1.39 (±3.16)	1.00 (±2.41)
**Activation**	6.14 (±1.94)	6.27 (±1.87)	5.73 (±2,07)	6.81 (±1.94)	6.96 (±1.87)	6.34 (±2.08)	0.67 (±2.13)	0.69 (±2.11)	0.61 (±2.20)
**Deactivation**	5.55 (±2.42)	5.62 (±2.51)	5.32 (±2,13)	4.26 (±2.38)	4.17 (±2.46)	4.51 (±2.10)	1.29 (±2.17)	1.45 (±2.01)	0.80 (±2.54)
**Excitement**	5.98 (±3.61)	6.02 (±3.72)	5.49 (±3,25)	3.43 (±3.24)	3.80 (±3.38)	2.27 (±2.42)	2.46 (±2.80)	2.21 (±2.75)	3.22 (±2.80)

**Table 3 brainsci-10-00376-t003:** Analytical statistics for the outcome variables; d = Cohen’s d and *p* = *P*-value.

Cohen’s d and *p*-Values for the Differences the Outcome Variables						
	d/All	d/Male	d/Female	*p*-Value/All	*p*-Value/Male	*p*-Value/Female
Elevated mood	1.1	1.0	1.3	<0.001	<0.001	<0.001
Low mood	0.3	0.4	0.4	<0.001	<0.001	<0.001
Activation	0.3	0.3	0.3	<0.001	<0.001	=0.0434
Deactivation	0.6	0.7	0.3	<0.001	<0.001	=0.0259
Excitement	0.9	0.8	1.2	<0.001	<0.001	<0.001

**Table 4 brainsci-10-00376-t004:** Results of the mixed ANOVA.

	Elevated Mood	Low Mood	Activation	Deactivation	Excitement
time	*F*(1.164) = 146.4 *p* < 0.001 η_p_^2^ = 0.47	*F*(1.164) = 18.8 *p* < 0.001 η_p_^2^ = 0.10	*F*(1.163) = 11.8 *p* = 0.001 η_p_^2^ = 0.07	*F*(1.163) = 33.1 *p* = 0.001 η_p_^2^ = 0.17	*F*(1.165) = 116.9 *p* = 0.001 η_p_^2^ = 0.42
sex T	*F*(1.164) = 0.07 *p* = 0.79 η_p_^2^ < 0.01	*F*(1.164) = 1.2 *p* = 0.28 η_p_^2^<0.01	*F*(1.163) = 0.3 *p* = 0.57 η_p_^2^<0.01	*F*(1.163) = 2.3 *p* = 0.13 η_p_^2^ = 0.01	*F*(1.165) = 3.3 *p* = 0.07 η_p_^2^<0.02
sex A	*F*(1.164) = 0.92 *p* = 0.34 η_p_^2^ < 0.01	*F*(1.164) = 2.6 *p* = 0.14 η_p_^2^ = 0.01	*F*(1,163) = 4.4 *p* = 0.04 η_p_^2^ = 0.03	*F*(1.163) = 0.0 *p* = 0.98 η_p_^2^ < 0.01	*F*(1.165) = 4.0 *p* < 0.05 η_p_^2^ = 0.02
time x sex T	*F*(1.164) = 2.1 *p* = 0.15 η_p_^2^ = 0.01	*F*(1.164) = 0.6 *p* = 0.43 η_p_^2^ = <0.01	*F*(1.163) = 0.0 *p* = 0.90 η_p_^2^ < 0.01	*F*(1.163) = 3.1 *p* = 0.08 η_p_^2^ < 0.02	*F*(1.165) = 0.4 *p* = 0.56 η_p_^2^ < 0.01
sex T x sex A	*F*(1.164) = 4.21 *p* = 0.04 η_p_^2^ < 0.01	*F*(1.164) = 0.5 *p* = 0.47 η_p_^2^ < 0.01	*F*(1.163) = 0.1 *p* = 0.73 η_p_^2^<0.01	*F*(1.163) = 1.0 *p* = 0.32 η_p_^2^<0.01	*F*(1.165) = 1.7 *p* = 0.20 η_p_^2^<0.01
time x sex T x sex A	*F*(1.164) < 0.01 *p* = 0.96 η_p_^2^ < 0.01	*F*(1.164) = 1.0 *p* = 0.31 η_p_^2^ < 0.01	*F*(1.163) < 0.0 *p* = 0.99 η_p_^2^ < 0.01	*F*(1.163) = 5.0 *p* = 0.03 η_p_^2^ = 0.03	*F*(1.165) = 0.7 *p* = 0.39 η_p_^2^ < 0.01

Time: main effect of time, sex A: effect of sex of the athlete, sex T: effect of sex of the therapist, time x sex T: two-way interaction of time and sex of the therapist, sex A x sex T: two-way interaction of sex of the therapist and sex of the athlete and time x sex A x sex T: three-way interaction of time, sex of the therapist and sex of the athlete. *F*: test of equality of variances, η_p_^2^: Partial Eta Squared.

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
