# Peer review of "Does the Therapist’s Sex Affect the Psychological Effects of Sports Massage?—A Quasi-Experimental Study"

_brainsci, 2020, doi:10.3390/brainsci10060376_

Round 1

Reviewer 1 Report

In row 218 `185 athletes are in cluded and received a massage

in row 260/261 `168 participants received a massage,

What about the difference of 17 persons?

Row 275 Table 3 exists twice;

Table 4 is not mentioned in the text.

Reviewer 2 Report

Overall, interesting unique study and design was appropriate.

No major changes needed as objectives and outcomes match (though title doesn't give the paper as much justice as other mood/emotional state changes were found). Methods appropriate. Conclusions drawn are appropriate.

– The dependent variables may have high collinearity, why was a MANOVA not used?

I do not really understand this question. In mixed ANOVAs, the partial etas are given; they are called manuscript "η²p". The values are given for the main effect (time, sex of the athlete, sex of the athlete) in each analysis.

There are several dependent variables (outcome) and these variables may be correlated with each other. If so what is there correlation and if the overlap Is big enough then A MANOVA should be run instead of an ANOVA.
